# Real-Time Receive-Forward NLOS Visible Light Communication System Based on Multiple Blue Micro-LED Nodes

Yuan Zhang [1], Zixian Wei [1], Zhaoming Wang [1] and H. Y. Fu [1,2,*]

1    Tsinghua Shenzhen International Graduate School, Tsinghua-Berkeley Shenzhen Institute (TBSI),
Tsinghua University, Shenzhen 518055, China; zhangyua20@mails.tsinghua.edu.cn (Y.Z.);
weizx17@tsinghua.org.cn (Z.W.); wangzm19@mails.tsinghua.edu.cn (Z.W.)
2    Peng Cheng Laboratory (PCL), Shenzhen 518055, China
\*    Correspondence: hyfu@sz.tsinghua.edu.cn

**Abstract:** A significant challenge of visible-light communication systems (VLC) is to overcome their limited converge area in non-line-of-sight (NLOS) transmission. To tackle this problem, for the first time, a real-time high-speed dual-hop VLC system based on blue micro-light-emitting diodes (micro-LED) is proposed and experimentally demonstrated. Benefiting from the advantage of high electrical-to-optical (E-O) bandwidth of the micro-LED, the frequency-response measurements show that the 3-dB modulation bandwidth of 2 m free-space single-hop link is 880 MHz, and the dual-hop system can reach to 715 MHz over a 4 m communication distance. We then investigated the communication performance of our proposed single-hop and dual-hop systems. The real-time waveforms are analyzed at different positions of the dual-hop link and eye diagrams at the receiving terminal are captured for evaluation. Furthermore, the bit error rate (BER) at the target node is measured. The results demonstrate that a 1.1 Gbps on-off keying (OOK) signal with a BER less than the forward-error-correction (FEC) limit could be achieved over a 4 m NLOS free-space link. This work shows that the proposed dual-hop system based on a micro-LED can meet the requirements for most indoor NLOS-transmission scenarios.

**Keywords:** visible-light communication (VLC); micro-LED; dual-hop transmission; non-line-of-sight (NLOS); receive-forward

## 1. Introduction

With the rapid deployment of the Internet of things (IoT) for smart cities and smart homes, the next generation of wireless communication systems are expected to have the potential to connect and support more devices, including computers, mobile phones, tablets, high-definition (HD) cameras, virtual reality (VR), etc. Communications between a large number of devices through traditional radio-frequency (RF) technology will cause serious congestion and interference problems. Visible-light communication (VLC) is a promising candidate to address the spectrum-scarcity issue, utilizing visible light as an information carrier with high security and anti-electromagnetic-interference (EMI) characteristics, simultaneously [1–3]. With the advantages of rich bandwidth resources and low cost, indoor VLC systems have great potential in solving the shortage of spectra and meeting the high data-rate requirements of systems other than 5G. However, due to the inescapable properties of lightwave transmission, line-of-sight (LOS) VLC cannot penetrate objects. VLC suffers from non-line-of-sight (NLOS) transmission constraints and a shadowing effect, which cannot guarantee stable and continuous communication, especially for indoor scenarios with random moving obstacles. Therefore, many methods are proposed to tackle this problem when the LOS link is not available.

Evolving from point-to-point VLC configuration, in order to improve the inherent issues of limited transmission distance or the link barrier, the multihop framework in VLC

systems has aroused intense research interest recently. In relay systems, the introduction of relay nodes provides more possibilities for optimizing the reliability of source-to-cell links. The concepts of relaying communication that could use certain light-emitting diode (LED) lamps as relays nodes were proposed and different indoor light sources can be deployed as relay nodes, such as ceilings, desks, and floor lights [4–7]. Relays between multiple VLC terminal nodes can effectively extend the transmission distance and reduce the dependence of LOS links, which have been experimentally demonstrated with improved performance [8]. The channel response of the VLC system with relay nodes based on an amplification and forwarding (AF) strategy was analyzed, and the relay auxiliary strategy in improving system reliability was simulated [9]. In addition, an experimental comparison of single-channel and relay-based VLC-link performance proved that the relayed VLC link can provide a higher data rate than the direct VLC link [10]. Until now, two types of transmission protocols have been proposed in relay communication, including decode-and-forward (DF) mode and AF mode [11]. Hyeong-ji Kim et al. proposed a multihop VLC system for offshore applications to overcome the limited coverage distance [12]. Omer Narmanlioglu et al. studied the performance of a full-duplex relay-auxiliary VLC system and proved that the full-duplex relay is superior to the half-duplex relay [13]. Alice Faisal et al. adopted a transmission-diversity scheme and showed that it can improve the signal-to-noise ratio (SNR) and reduce outage probability [14]. Kuan Ye et al. investigated the performance of dual-hop underwater optical wireless communication (UOWC) systems with simultaneous lightwave information and power transmission, and numerical results showed that using relay nodes can improve performance [15]. Overall, the relay schemes can be divided into two types: passive relay and active relay. Passive relays currently have some proposed solutions such as intelligent reflecting surfaces (IRS), different reflecting materials, and concave mirrors. These works are mainly for beam forming and beam steering of the emitting ray. Using a concave mirror cannot change the direction of the beam very well, which is usually used to increase the intensity of the beam in point-to-point channels. Another way, the active relay, we will mention below.

The above-mentioned works are all based on simulations without experimental results. In recent years, some experiments have been also carried out based on multihop VLC. For example, the demonstration of a relay-assisted VLC system based on Multiband carrierless amplitude and phase modulation (M-CAP) was carried out with a data rate of 10 Mbps [10]. A bidirectional multihop VLC was used to monitor large-area indoor fine particles. The distance between two nodes can reach 13.5 m but the data rate is only 115.2 Kbps with on-off keying (OOK) format, which is unsuitable in high-speed scenarios [16]. Elizabeth Eso et al. proposed a relay-assisted vehicle-mounted VLC network based on experimental measurements and presented the eye diagrams from 250 Kbps to 500 Kbps [17]. In addition, the performance of a multihop VLC system can be also tested in a real outdoor environment; however, the work only gave the relationship between the bit-error rate (BER) and average transmission power, which inadequately represents real-time communication [18]. Nonetheless, relatively high-speed multihop experiments are rarely reported. For example, a channel-aware adaptive physical-layer network-coding scheme based on adaptive-loading orthogonal frequency-division multiplexing (OFDM) was proposed, which can double the throughput of a relay-assisted VLC network [19]. A VLC system integrated with vertical-cavity surface-emitting laser (VCSEL) and LED was demonstrated indoors using a DF relay scheme with a data rate of 650 Mbps over a 4 m link [20].

Overall, most reported works on multihop free-space VLC systems are usually based on numerical simulations or only for low-speed scenarios. The data rate of the multihop system was limited significantly by the electrical-to-optical (E-O) bandwidth of the transmitter. Using micro-light-emitting diodes (micro-LEDs) as a transmitter has great potential for high-speed multihop VLC implementation due to their high E-O bandwidth characteristics [21,22]. In Ref. [23], a high-bandwidth micro-LED with a self-assembled nanostructure InGaN wetting layer was designed and fabricated, which shows great po-

tential in point-to-point VLC applications. In addition, the maximum achievable data rate of 4 Gbps has been experimentally demonstrated by using OFDM. In order to overcome the limited communication distance and increase the data rate in the real-time NLOS VLC systems, a multihop structure combined with high-bandwidth micro-LED was adopted in this work. Incidentally, micro-LEDs also have great potential in the field of ultraviolet communication (UVC). Both NLOS and relatively low-speed problems can also be solved in ultraviolet (UV) micro-LED-based UVC systems. Recently, an AlGaN-based deep UV micro-LED emitting at 275 nm was proposed with the 3-dB E-O bandwidth of 380 MHz [24]. In addition, high-speed UVC based on a 276.8 nm UV micro-LED with a 3-dB E-O bandwidth of 452.53 MHz was experimentally achieved. A UVC link over 3 m with a data rate of 0.82 Gbps was presented [25]. However, ultraviolet radiation is not suitable for indoor scenarios resulting from its characteristics, which can be harmful and cause different degrees of damage to the human body. Moreover, our proposed dual-hop VLC system based on a blue micro-LED showed a higher communication-data rate compared with UVC. Therefore, UVC is not suitable for solving NLOS problems for indoor applications, and our dual-hop VLC system is a promising solution to tackle high-speed NLOS problems indoors.

In this work, in order to explore the potential of the multihop structure, a dual-hop AF relay-based VLC system with a wider coverage range and a higher data rate is proposed. We focus on the application of VLC in an indoor environment, by fully routing data between VLC-based nodes and providing a Gbps communication rate. Two 488 nm blue micro-LEDs with 880 MHz E-O bandwidth were designed and then used as the transmitter in two nodes. The modulation characteristics of the micro-LED are the key reason to enable the high-speed system than other works. After a total 4 m free-space transmission, the qualitative analysis of VLC links was carried out by frequency response and real-time BER measurements with a simple OOK-modulation scheme. After passing through the relay node, the NLOS problem could be solved and the communication distance was then extended; the consequent inevitable cost was a reduction in modulation bandwidth from 880 to 715 MHz. This is mainly due to the noise introduced by AF and the long-distance attenuation. On the whole, the communication system also remained high data rates and a slightly reduced modulation bandwidth, while the cost of the dual-hop VLC system was acceptable. In addition, we also studied the influence of distance on the dual-hop system. To the best of our knowledge, this is the first real-time dual-hop link based on micro-LEDs. The results show that our proposed dual-hop VLC system with relay infrastructure can support future high-speed optical wireless networks by extending the coverage area for indoor NLOS-transmission applications.

The rest of this paper is organized as follows: Section 2 describes the concept and channel-model dual-hop VLC system based on micro-LEDs. Section 3 shows the experimental setup for the proposed dual-hop VLC system. System communication performance including the modulation bandwidth and the BER results with real-time is also given in this section. Finally, we conclude the investigation in Section 4.

## 2. Concept, System Setups, and Methods

In Figure 1, a relaying VLC system based on multiple micro-LEDs is investigated, which consists of the source node, the relay node, and the end-user. The corresponding experimental setup of dual-hop system is shown in Figure 2. Experimental parameters based on Hop 1 and Hop 2 are summarized in Table 1. The proposed framework can be applied in indoor VLC scenarios, which means many high-speed applications including virtual reality, real-time HD communication, second-level download, and real-time cloud storage can be realized. High-speed data transmission not only improves the ultimate sensory experience for users but also creates greater value for operators. However, the transmission system may be limited by NLOS due to the characteristics of VLC. Therefore, we consider the downlink transmission of a dual-hop system, consisting of two micro-LEDs and a single end-user. The system concept is shown in Figure 1a. As illustrated in Figure 1a, the link between the

first transmitter (Tx.1) and the second receiver (Rx.2) suffers NLOS transmission constraints due to the obstacle. The use of a relay node composed of Rx.1 and Tx.2 is an alternative method to solve this NLOS problem. We used a micro-LED as the light source for the dual-hop VLC system; therefore, the transmitter can be considered as a monochromatic-point light source with Lambertian radiation pattern. The VLC system is based on the AF relay method. Among various relay strategies based on VLC, we chose AF relay as our relay node, which is very simple but will inevitably synchronously cause amplification of noise. In our dual-hop VLC system, channel-intensity modulation with direct detection (IM/DD) is the preferred choice, which can be modeled as a base-band linear system. At the end-user receiver, the noise is composed of larger shot noise and thermal noise due to the introduction of additional relay node. In a multi-relay VLC system that implements communication, there is a positive correlation between the communication-performance deterioration of the end-user receiver and the number of introduced nodes.

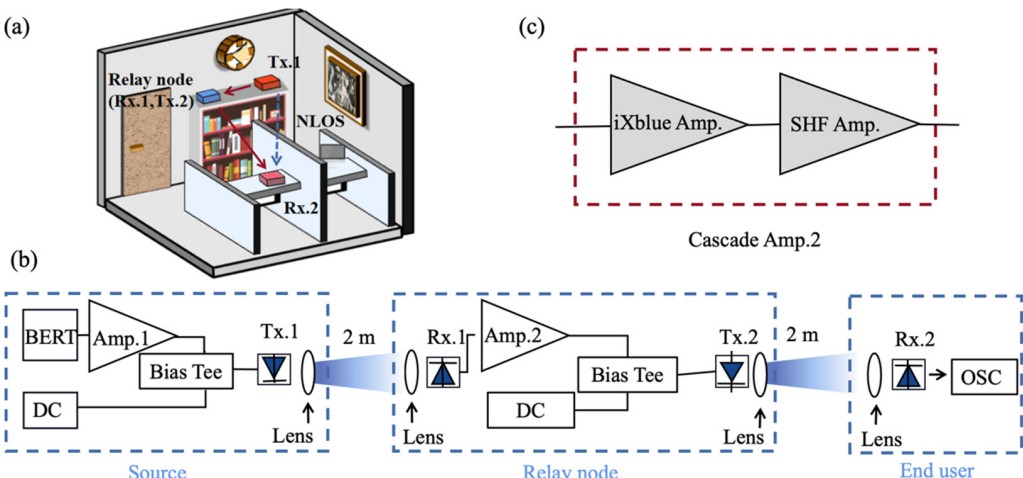

**Figure 1.** (**a**) Schematic diagram of dual-hop VLC-link setup. (**b**) Diagram of dual-hop VLC-link setup. (**c**) The cascade structure of Amp.2 with detailed order and type. (Tx.: Transmitter; Rx.: Receiver; Amp.: Amplifier; BERT: Bit-error-rate tester; DC: Direct current).

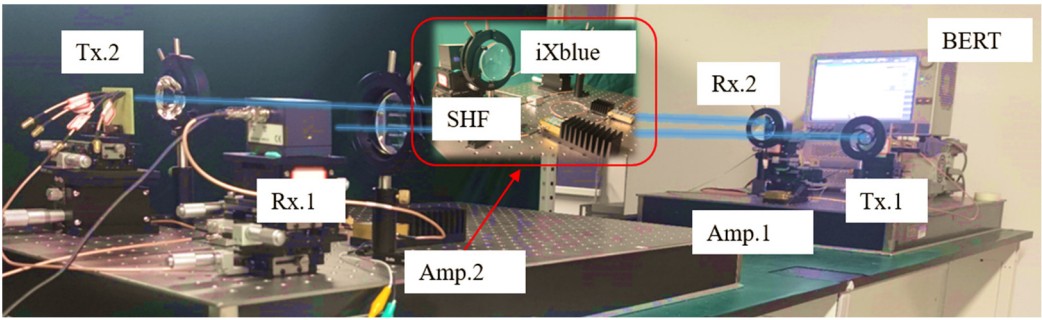

**Figure 2.** Experimental setup of dual-hop VLC link. (Inset: Cascade Amp.2).

The setup for the dual-hop VLC system is depicted in Figure 1b. A long pseudo-random binary sequence with a length of $2^{31}-1$ is modulated by OOK format and generated from bit-error-rate tester (BERT, MP2100B, Anritsu), the system also used to measure BER performance. Before implementing the intensity modulation of micro-LED, the signal is amplified first and superimposed with direct current (DC) by a bias-tee. After 2 m free-space transmission, the optical signal is received by Rx.1. In this experiment, Rx.1 and Rx.2 are two avalanche photodiode (APD, APD210, Thorlabs) modules. Following transmission over the first optical link, the AF relay scheme is used. At the relay node, Rx.1 is used to convert the received optical signals into electrical signals and provide high gain. In order to ensure the signal quality and

adjust the voltage dynamically, Amp.2 is composed of two amplifiers (SHF, Mini-circuits, and DR-AN-20-MO, iXblue), as shown in Figure 2. To obtain the optimum voltage and modulation depth for relay signals, we can traverse different values to select the appropriate voltage corresponding to the lowest BER, specific voltage values can be seen in Table 1. Similarly, Rx.2 can detect the optical signal transmitted from the Tx.2 after a 2 m distance in another direction. Finally, to evaluate communication performance, a real-time oscilloscope (RTO, DPO75902SX, Tektronix) is used to record the signal after dual-hop structure, and then analyze the signal quality and capture the eye diagrams. The modulation bandwidth of the VLC system is obtained by a vector network analyzer (N5227A, Keysight) which connects with the transmitter and the receiver side. BERT is used to transmit and receive the signals in order to measure BER at different data rates.

**Table 1.** Experimental parameters based on Hop 1 and Hop 2.

| Parameter | Hop 1 | Hop 2 |
|---|---|---|
| Transmitter | 100 μm micro-LED | 100 μm micro-LED |
| Output optical power | 67 μW | 63 μW |
| Receiver | APD210 (1 GHz) | APD210 (1 GHz) |
| The length of PRBS | $2^{31}-1$ | $2^{31}-1$ |
| Modulation format | OOK | OOK |
| Bias voltage | 4.48 V | 4.49 V |
| Modulation depth (Vpp) | 5.3 V | 5.2 V |
| Amplifier | SHF | SHF iXblue |
| Distance | 2 m | 4 m |

## 3. Experimental Results and Discussions

### 3.1. Device Characteristics

Detailed electrical and optical properties of the packaged 100-μm blue micro-LED are measured, as shown in Figures 3 and 4.

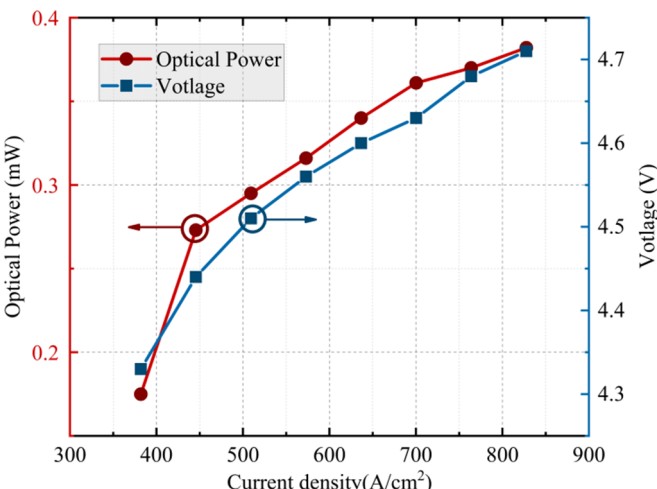

**Figure 3.** Current density vs. voltage (*J-V*) and current density vs. optical power density (JL) curves of the package 100-μm micro-LED.

Figure 3 shows the current density and voltage (*J-V*) and current density and optical power density (*J-L*) of the package 100-μm micro-LED. *J-V* characteristics were measured by a DC supply instrument under different current densities (RIGOL, DP800). JL characteristics were measured using a calibrated silicon photodiode detector (Thorlabs, PM100D). As shown in Figure 3, the linear region of the micro-LED is small. However, the nonlinear part in the subsequent experiment can be used to improve the SNR and the communication performance. The optimal value of DC and modulation depth of the system can be found by traversing.

As shown in Figure 4, the optical spectrum of the micro-LED is measured at various driving current densities, which can observe that the emission peak is around 488 nm. As the current density changes from 509.28 $A/cm^2$ to 891.24 $A/cm^2$, it can be seen that there is no obvious red/blue shift. Meanwhile, the intensity peaks will all significantly rise as the current density increases.

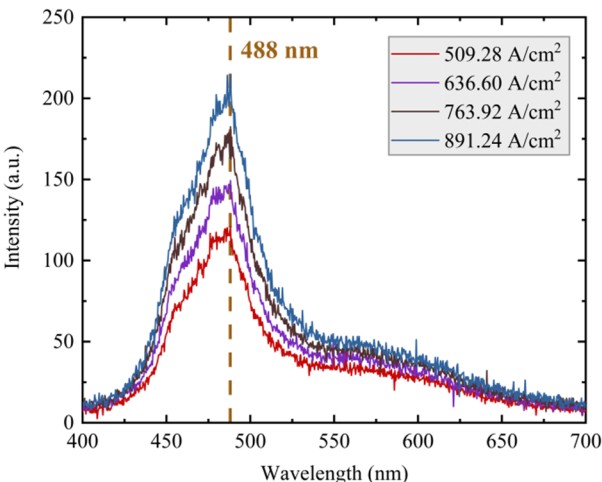

**Figure 4.** The spectrum of the package 100-μm micro-LED.

### 3.2. Analysis of Signal Waveform Quality

To qualitatively analyze the signal distortion in this dual-hop data-transmission system, a BERT is used to send a $2^{31}-1$ pseudo-random binary sequence (PRBS) at 1 Gbps and a real-time oscilloscope is used to capture the waveform, simultaneously. Figure 5 shows the signals at the test points before Tx.1, after Rx.1, Amp.2, and Rx.2, respectively. Furthermore, the peak-to-peak-voltage (Vpp) is also noted. Figure 5a,b shows that the OOK signal suffers distortion and overlaps introduced noise as it passes through the 2 m link. To obtain an ideal signal strength to drive Tx.2, the electrical signal generated by Rx.1 is amplified. The amplifier (DR-AN-20-MO, iXblue, BesanCon, France) provides a tunable gain of 22 dB, which amplifies the electrical signal from 137.6 mV to 4.8 V. This voltage serves as the optimum modulation depth. The waveform is closer to a square wave after the amplifier, mainly because of the influence of oscilloscope noise superimposed on the final waveform instead of the improvement of the signal-to-noise ratio (SNR), as shown in Figure 5b,c. Then, the signal passes through the second 2-m link. Figure 5c,d shows that the OOK signal suffers similar distortion and overlaps introduced noise as it passes through the second 2-m link.

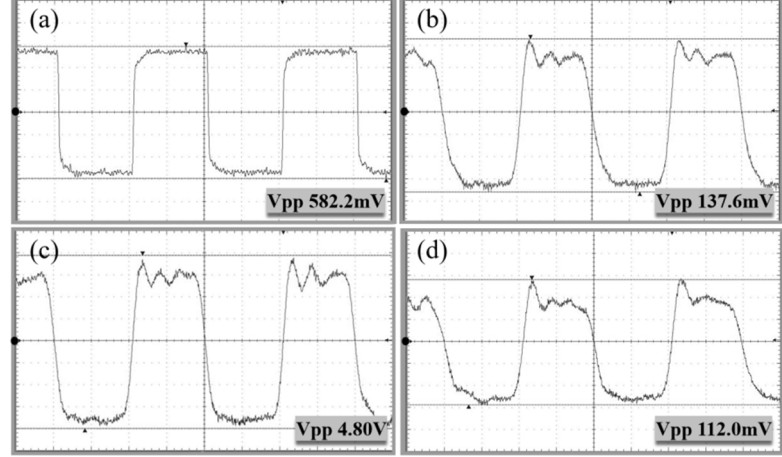

**Figure 5.** Signal observed on oscilloscope after (**a**) Tx.1, (**b**) Rx.1, (**c**) amplifier, (**d**) Rx.2.

### 3.3. Communication-Performance Evaluations

In this section, an evaluation of the communication performance of the dual-hop system is given. In order to better understand the influence of the communication performance of the dual-hop system, a comparison is made with the single-hop system. The frequency response and BER performance were experimentally tested.

The normalized frequency response of the micro-LED-based VLC system at different links is shown in Figure 6, which illustrates the modulation-bandwidth characteristics under different situations. Without any hardware-equalization technique, it can reach 880 MHz in the single-hop link, which presents this blue micro-LED has great potential as high-speed communication application. Moreover, the values of 3-dB modulation bandwidth decrease to 715 MHz with the relay node. It is easy to see that modulation bandwidth only slightly decreases after the relay nodes and longer distance transmission.

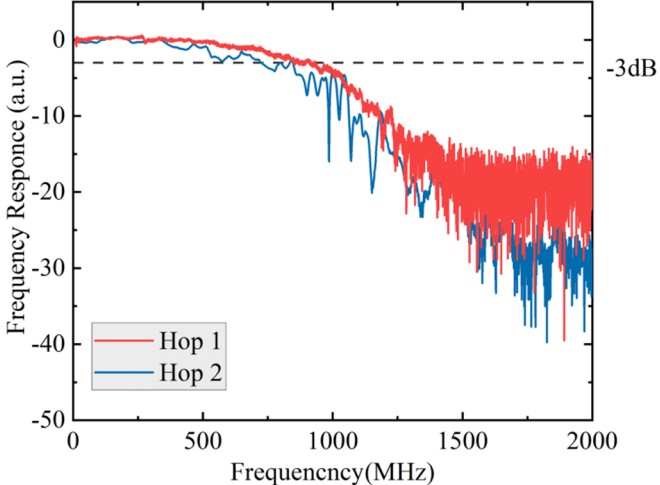

**Figure 6.** The comparison of measured normalized frequency response from Hop 1 and Hop 2.

Figures 7 and 8 indicate the results of measured BER performance, as well as the obtained insert of it, which represents the corresponding eye diagrams when the data rate is varied. The PRBS pattern length of the OOK signal is set as $2^{31}-1$. As illustrated in Figure 6, the different OOK modulations of 1.0 Gbps to 2.1 Gbps are applied on Tx.1. The maximum data rate is 2.0 Gbps within the forward error correction (FEC) limit ($3.8 \times 10^{-3}$), which can fully demonstrate the great potential of the micro-LED used in the high-speed VLC communication system. As a result, the maximum data of the first blue micro-LED can reach 2 Gbps in 2 m free-space transmission at the FEC limit. Meanwhile, through the same 2 m free-space channel and lens, the values of the received optical power are also measured, which remain relatively the same: 67 µW in Figure 7 while 63 µW in Figure 8.

Then, the maximum modulated data of Tx.2 is likewise performed. In the experiment, the different OOK modulations of 500 Mbps to 1.3 Gbps are applied on the second blue micro-LED. The maximum data rate corresponding to the FEC limit is 1.1 Gbps, which is smaller than the single-hop link. After the AF method, the transmitted optical power of the relay node is almost the same. The change of Vpp also confirms this, which is from 137.6 mV to 112.0 mV. Therefore, for single-hop and dual-hop links, 2.0 Gbps and 1.1 Gbps are achievable within the FEC limit, respectively.

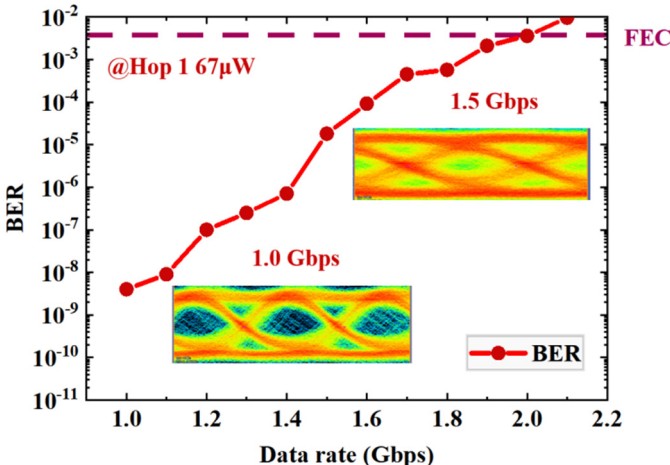

**Figure 7.** Measurement of real-time BER with OOK format from Hop 1. (Inset: the corresponding eye diagrams at the data rates of 1.0 Gbps and 1.5 Gbps).

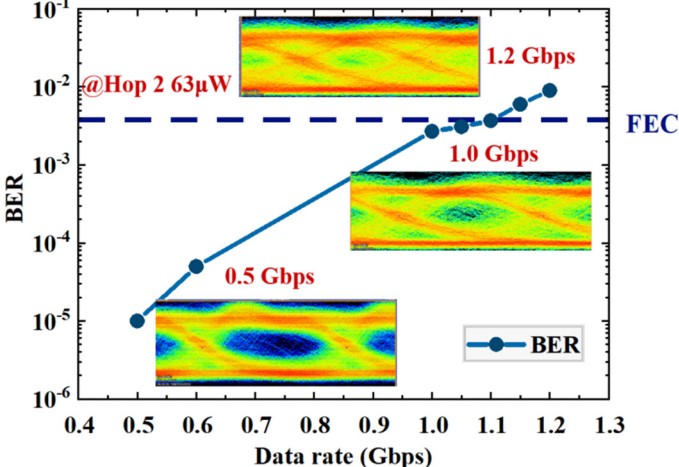

**Figure 8.** Measurement of real-time BER with OOK format from Hop 2. (Inset: the corresponding eye diagrams at the data rates of 500 Mbps, 1.0 Gbps, and 1.2 Gbps).

In order to be more intuitive, Table 2 summarizes some measured indicators related to communication performance. Under the premise of the extended distance of the dual-hop VLC system which increases from 2 m to 4 m, it is noticeable that some of the communication performance is changed. The value of Vpp goes down from 137.6 mV to 112.0 mV, which is relatively the same due to the AF method and the same bias current. However, the AF method will inevitably introduce synchronous amplification of noise; the noise and long-distance attenuation represent a greater impact in the dual-hop link following an inevitable reduction in the modulation bandwidth and the maximum data rate. There is an admissible decrease in modulation bandwidth between 880 MHz and 715 MHz. Moreover, the data rate stabilizes at the order of Gbps and the relay node in the real-time dual-hop VLC system expands the scope of the limited area, tackling the NLOS problem effectively. Therefore, the cost of the relay node is acceptable. To the best of our knowledge, this is the first experimental demonstration of a real-time dual-hop link based on a micro-LED that could provide a potential solution to tackle the NLOS problem of indoor VLC. In this work, the modulation signal with an OOK format is used to complete the experimental verification of our proposed dual-hop VLC system based on blue micro-LEDs. In future work, higher-order modulation formats can be considered to obtain higher data rates.

**Table 2.** Communication-performance comparison from Hop 1 and Hop 2.

| Link | Distance | Vpp | Bandwidth | Data Rate (FEC) |
|---|---|---|---|---|
| Hop 1 | 2.0 m | 137.6 mV | 880 MHz | 2.0 Gbps |
| Hop 2 | 4.0 m | 112.0 mV | 715 MHz | 1.1 Gbps |

### 3.4. The Impact of Transmission Distance

To further understand the impact of transmission distance and prove that our proposed system can be applied to more scenarios, the measurement results at different distances are discussed in this section. Except for the change of communication distance, the system parameters are consistent with Table 1.

First, to have a better understanding of the distance impact to the received signal based on the micro-LED, we can measure the corresponding received optical power and Vpp in the free-space transmission of 2 to 4 m. As we can see from Figure 9, the received optical power is 16, 10, 8 μW at 2, 3, 4 m. The Vpp of the received signal is proportional to the received optical power. At the data rate of 1.5 Gbps, the eye diagrams of 2 m and 4 m are given, respectively. It can be seen that the communication performance of a single-hop system will deteriorate with the increase in transmission distance.

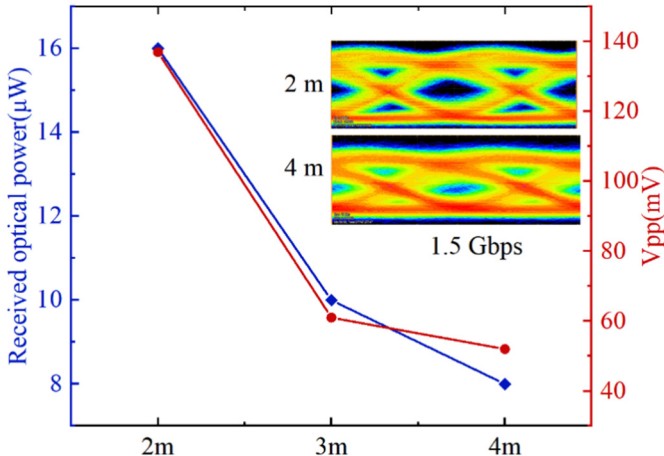

**Figure 9.** Measured received power and Vpp under the different free-space transmission lengths of 2 m to 4 m (Inset: the eye diagram at the data rate of 1.5 Gbps at 2 m and 4 m).

Then, the relationship between distance and data rate is analyzed. The comparison between Hop 1 and Hop 2 at different communication distances is given, which is shown in Figure 10. In the single-hop system, the data rates within FEC are 2 Gbps in the cases of 2 m and 3 m transmission distance, and the BER performance of 3 m is slightly increased with the increase in distance. When the distance increases to 4 m, the maximum data rate of the single-hop system decreases to 1.6 Gbps. In the dual-hop system, the data rates within FEC are 1.1 Gbps, 1.1 Gbps, and 1 Gbps at the length of 2 m to 4 m. It can be seen that when the single-hop communication performance is not limited by distance, the dual-hop communication performance also remains relatively stable with the increase in distance, which can further prove the potential application indoors.

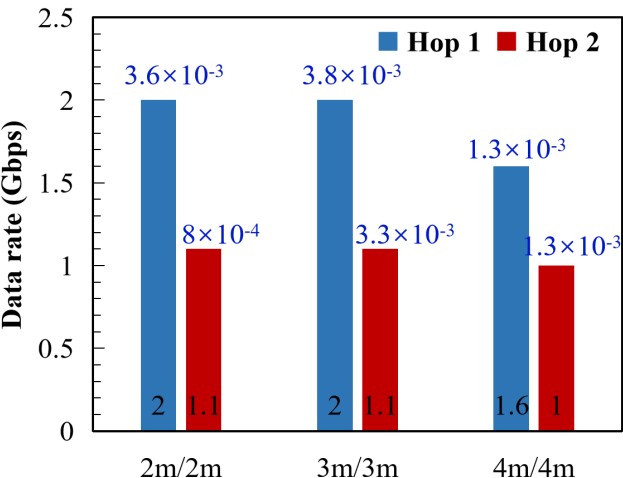

**Figure 10.** Measurement of maximum data rate within FEC limit at different lengths (The blue font on the histogram represents the BER for the corresponding case).

## 4. Conclusions

For the indoor environment, the LOS VLC system could not always have been guaranteed due to many effects, but it can greatly influence the communication quality. To address this unavoidable problem and have high-speed communication simultaneously, a dual-hop VLC system based on a blue micro-LED was proposed and experimentally demonstrated. The deployment of relay nodes can ensure high-speed communication indoors and solve the NLOS problem. To the best of our knowledge, this was the first time that a dual-hop VLC link was based on a micro-LED. In addition, in order to better analyze the cost and benefit of the relay node, a comparison between the single-hop and the dual-hop link was presented. On the premise of increasing the communication distance from 2 m to 4 m, there was an admissible decrease in modulation bandwidth between 880 MHz and 715 MHz. Meanwhile, the data rate corresponding to the FEC limit remained in the order of Gbps. The result demonstrated a dual-hop VLC system based on blue micro-LEDs with a data rate up to 1.1 Gbps through simple OOK modulation over a 4 m NLOS free-space link, with the help of a low-cost AF relay. In addition, when the overall link distance increases from 4 m to 8 m, the dual-hop system is still capable of high-speed communication. The data rate within the FEC limit at 8 m is 1 Gbps. The available 8 m transmission distance exceeded the LOS distance for the indoor LOS link, which can be applied to more indoor scenarios. Experimental results confirmed that the dual-hop VLC system based on micro-LEDs can tackle critical NLOS problems in VLC systems and is sufficient for high-speed indoor application in the future.

**Author Contributions:** Conceptualization, Z.W. (Zixian Wei) and Y.Z.; methodology, Z.W. (Zixian Wei); validation, Z.W. (Zixian Wei) and Y.Z.; formal analysis, Y.Z.; investigation, Y.Z.; resources, Z.W. (Zixian Wei) and Zhang, Y; data curation, Z.W. (Zixian Wei) and Z.W. (Zhaoming Wang); writing—original draft preparation, Y.Z., Z.W. (Zixian Wei) and Z.W. (Zhaoming Wang); writing—review and editing, H.Y.F.; visualization, Y.Z.; supervision, H.Y.F.; project administration, H.Y.F.; funding acquisition, H.Y.F. All authors have read and agreed to the published version of the manuscript.

**Funding:** This work is supported by Shenzhen Technology and Innovation Council (WDZC202008 20160650001).

**Institutional Review Board Statement:** Not applicable.

**Acknowledgments:** The authors thank their colleagues Lei Wang, and Lai Wang from Department of Electronic Engineering, Tsinghua University, for their kind help toward research work.

**Conflicts of Interest:** The authors declare no conflict of interest.

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
