# Peer review of "Real-Time Receive-Forward NLOS Visible Light Communication System Based on Multiple Blue Micro-LED Nodes"

_photonics, doi:10.3390/photonics9040211_

Round 1

Reviewer 1 Report

The draft reports the description of a novel architecture for visible light communication (VLC) with Receive-forward NLOS nodes based on micro-LED. The draft is mainly focused on the development of single-hop and dual-hop systems. Overall, the draft is informative and the topic is attractive to the VLC audience. The organization of the draft is clear and easy to follow. However, in my opinion, before it can be published, there are some aspects that the Authors should clarify which are listed below:

  1. In the introduction, the transmission protocol amplification and forwarding (AF) is mentioned. However, the following listed references [12]-[15] did not mention this protocol. Had this works involved the AF?
  2. The context of ultraviolet communication (UVC) in introduction part should be rewrite, which confuse reader now.
  3. The tense in the literature review of the introduction part is inconsistent and should be standardized to the past tense.
  4. In the experimental results and discussions section, there are not figure 6(a) and figure 6(b). However, figure 6(a) and figure 6(b) are mentioned in page 8. Authors should double check it.

Author Response

Response to reviewer 1 comments (Photonics-1642815)

Overall Comments:

The draft reports the description of a novel architecture for visible light communication (VLC) with Receive-forward NLOS nodes based on micro-LED. The draft is mainly focused on the development of single-hop and dual-hop systems. Overall, the draft is informative and the topic is attractive to the VLC audience. The organization of the draft is clear and easy to follow. However, in my opinion, before it can be published, there are some aspects that the Authors should clarify which are listed below:

Response:

Thanks for the comments. We are very grateful to the reviewer for providing valuable suggestions on our manuscript. We appreciate your appraise comments above and attach great importance to your proposed comments. And we would like to thank him/her for the raised questions to further improve this manuscript to be ready for publication in MDPI Photonics.

Comment 1:

In the introduction, the transmission protocol amplification and forwarding (AF) is mentioned. However, the following listed references [12]- [15] did not mention this protocol. Had this works involved the AF?

Response:

I am very grateful to reviewer 1 for pointing out this problem. After re-reading it, I found that this paragraph may lead to some misunderstanding as I will elaborate on the AF relay model next. In fact, references [12]- [15] are intended to illustrate relevant research related to relaying, not just the AF method. AF relay links suffer from inevitable noise accumulation in the transmission range but have the advantage of low cost and relatively simple structure. This sentence in the manuscript is just used to explain why we use the AF relay method for experimental demonstration here, but the synonym of this sentence is also mentioned in the next part. Under full consideration, we have deleted the sentence to make them easier to read and not confuse the reader.

Following the valuable comment, we have rewritten this paragraph in the revised manuscript as follows:

Until now, two types of transmission protocols have been proposed in relay communication, including decode-and-forward (DF) mode and AF mode [11]. Hyeong-ji Kim et al. proposed a multi-hop VLC system for offshore applications to overcome the limited coverage distance [12].

Comment 2:

The context of ultraviolet communication (UVC) in the introduction part should be rewritten, which confuses the reader now.

Response: Thanks for the comments. Ultraviolet communication has the advantages of low solar background noise, NLOS, and good security. The strong scattering property of the ultraviolet band makes it an ideal choice for NLOS optical communication and a typical solution to the NLOS problem. Therefore, ultraviolet communication is mentioned here but I'm really sorry that I didn't explain it clearly. To better illustrate UVC in the introduction, we have rewritten this paragraph to reduce the reader's confusion about the UVC in the introduction in the revised manuscript as follows:

Therefore, UVC is not suitable for solving NLOS problems for indoor applications, our dual-hop VLC system is a promising solution to tackle high-speed NLOS problems indoors.

Comment 3:

The tense in the literature review of the introduction part is inconsistent and should be standardized to the past tense.

Response: Thanks for the comments and pointing it out! We wrote those sentences incorrectly. We will change it in the revised version. We have proofread the paper and corrected such issues. Consequently, we have standardized the tense in the literature review of the introduction part and the conclusion part, please refer to it. In addition, we have carefully examined other details about the introduction, which have been highlighted in our revised manuscript.

Comment 4:

In the experimental results and discussions section, there is no figure 6(a) and figure 6(b). However, figure 6(a) and figure 6(b) are mentioned on page 8. The authors should double-check it.

Response: Thanks for the comments. We apologize for these mistakes and we have corrected them in the revised manuscript. In addition, we also made a careful examination of other parts of the article and made two revisions.

The waveform is closer to a square wave after the amplifier, mainly because of the influence of oscilloscope noise superimposed on the final waveform instead of the improvement of signal-to-noise ratio (SNR), as shown in Figure 5(b) and Figure 5(c). Then, the signal passes through the second 2-m link. Figure 5(c) and Figure 5(d) show that the OOK signal suffers similar distortion and overlaps introduced noise as it passes through the second 2-m link.

Meanwhile, through the same 2-m free space channel and lens, the values of the received optical power are also measured, which remain relatively the same: 67 μW in Figure 7 while 63 μW in Figure 8.

Reviewer 2 Report

The article is interesting but requires some additions:

  1. Paragraph 2 of the article should be extended. Please present in detail the proposed VLC dual hop system.
  2. Please add a photo showing the experimental stand on which you performed the tests.
  3. What happens when RX.2 is mobile?  If you can, it would be very interesting to present this test as well.

Author Response

Please check the attachment: the part of Reviewer 2. Thanks!

Reviewer 3 Report

The authors demonstrate relay-based VLC system with blue micro-LED as the light source to tackle the NLOS problem. The concept is very similar to that in Ref[10] except for a much higher data rate enabled by the blue micro-LED light source. The data rate, waveform distortion and BER are analyzed to show that the proposed VLC system meets the criteria of common indoor applications. However, the reviewer has several comments:

  1. The originality of the work should be further explained since the relay-based VLC is experimentally demonstrated. Comparison between the proposed system in this work and previous works should be added to clarify the source of the data rate improvement: if it is simply due to a different light source or in the electrical components.
  2. Since the input and output signals of the relay node are both directional visible lights, a concave mirror might fulfill part of the relay’s role. Therefore, a comparison between the proposed solution and other NLOS solutions would be beneficial to demonstrate the advantage of the relay-based system.
  3. Another advantage of the relay node besides the NLOS problem is the extended communication distance from 2 meters to 4 meters, at the cost of a reduced bandwidth. However, systematic study of the data rate vs distance relationship is missing, which is required to evaluate the impact of the relay node on the communication distance.

Author Response

Please check the attachment: the part of Reviewer 3. Thanks!

Round 2

Reviewer 3 Report

The authors have addressed most of the reviewers' comments and made improvement to the manuscript. It is now suitable to be published.